# The Role of Voltage-Dependent Anion Channel in Mitochondrial Dysfunction and Human Disease

**DOI:** 10.3390/cells10071737

**Published:** 2021-07-09

**Authors:** Joyce T. Varughese, Susan K. Buchanan, Ashley S. Pitt

**Affiliations:** 1Laboratory of Molecular Biology, National Institute of Diabetes and Digestive and Kidney Diseases, National Institutes of Health, Bethesda, MD 20892, USA; joyce.varughese@nih.gov (J.T.V.); susan.buchanan2@nih.gov (S.K.B.); 2Department of Biology, The Johns Hopkins University, Baltimore, MD 21218, USA

**Keywords:** voltage-dependent anion channel (VDAC), outer mitochondrial membrane, mitochondria-associated membranes, ATP transport, calcium homeostasis, disease

## Abstract

The voltage-dependent anion channel (VDAC) is a β-barrel membrane protein located in the outer mitochondrial membrane (OMM). VDAC has two conductance states: an open anion selective state, and a closed and slightly cation-selective state. VDAC conductance states play major roles in regulating permeability of ATP/ADP, regulation of calcium homeostasis, calcium flux within ER-mitochondria contact sites, and apoptotic signaling events. Three reported structures of VDAC provide information on the VDAC open state via X-ray crystallography and nuclear magnetic resonance (NMR). Together, these structures provide insight on how VDAC aids metabolite transport. The interaction partners of VDAC, together with the permeability of the pore, affect the molecular pathology of diseases including Parkinson’s disease (PD), Friedreich’s ataxia (FA), lupus, and cancer. To fully address the molecular role of VDAC in disease pathology, major questions must be answered on the structural conformers of VDAC. For example, further information is needed on the structure of the closed state, how binding partners or membrane potential could lead to the open/closed states, the function and mobility of the N-terminal α-helical domain of VDAC, and the physiological role of VDAC oligomers. This review covers our current understanding of the various states of VDAC, VDAC interaction partners, and the roles they play in mitochondrial regulation pertaining to human diseases.

## 1. Introduction

Mitochondria have two membranes: the outer mitochondrial membrane (OMM) and the inner mitochondrial membrane (IMM). The proteins contained within these membranes enable mitochondria to carry out essential functions including energy production, calcium homeostasis, protein quality control, protein import, signaling, fission, and fusion. The ways that metabolites and molecules traverse membranes are important for understanding many human diseases, especially chronic and age-related diseases [1].

The OMM contains gated porins that permit passage of peptides, metabolites, and ions. The voltage-dependent anion channel (VDAC) is an example of a gated porin that allows the passage of metabolites and ions, and it is one of the most abundant proteins within the OMM [2]. VDAC interacts with other OMM proteins that are involved in mitochondrial protein transport. As a transporter for metabolites and ions, VDAC is involved in mitochondrial outer membrane permeabilization (MOMP) and in establishing a possible OMM membrane potential [3,4,5]. MOMP and membrane potential are vital in the mitochondrial function. Improper regulation of OMM membrane potential maintenance as well as the permeability of the membrane can lead to apoptosis [6].

VDAC has two states that allow for the selective passage of ions and metabolites: open and closed. The open state is primarily permeable to anions and metabolites, the most important of which is ATP. Three structures of mammalian VDAC orthologs in the open state have been solved via NMR and X-ray crystallography [7,8,9]. There are currently no structures of apo-VDAC in the voltage closed state. A structure of VDAC with NADH bound, in the ligand-bound closed state, solved by NMR, does exist [10]. Biochemical and biophysical data suggest that the voltage closed state occurs at higher membrane potentials is more permeable to cations and is impermeable to ATP. It is thought that multiple closed state conformations exist of VDAC (Table 1) [11]. The structural changes necessary for closing of the barrel still remain unknown.

While VDAC can exist in a closed state at high membrane potential, VDAC permeability can also be modulated via binding partners, as seen with hexokinase, tubulin, or alpha-synuclein which bind to VDAC and block the pore, leading to a reversible closed state [12,13]. Studies have observed a flickering of closed to open states at membrane potentials as low as 10 mV upon the addition of tubulin [14]. This reversible closure differs from the high voltage-dependent closure which closes at voltages around 50 mV and only relaxes to the open state when the voltage reaches 0 mV [14]. Another key difference seen between the closed state due to ligand binding versus the high voltage-dependent closure is that there is only one closed state observed. In comparison, multiple closed states were observed for the high voltage dependent-closure of VDAC. In addition, NADH, which is a metabolite that can translocate through the pore, can also bind to the wall of VDAC and can block the pore [10]. Although studied for many decades, VDAC’s exact relationship between its gating and structural conformations is still debated. Understanding this relationship is vital for exploiting the therapeutic potential of VDAC, for example in oligomers or within mitochondrial-associated membranes (MAMs) [6,15]. Further study is needed to understand whether structural conformational changes are required for gating.

This review will explore the structure, interactions, and binding partners of VDAC that are key to understanding VDAC’s role in disease pathogenesis, as well as how VDAC can be selectively targeted for therapeutics.

## 2. General Role of VDAC

VDAC is a β-barrel membrane protein located in the OMM that regulates the entry and exit of key metabolites using a voltage-sensing mechanism and via ligand binding in physiologically relevant conditions. At low voltages, VDAC is predominantly in an open state and engages in preferential transport of anionic metabolites, while higher cationic permeability is seen at higher voltages [16,17]. Examples of water-soluble metabolites and ions regulated by VDAC include calcium, ATP, ADP, NADH/NAD+, nucleotides, and citrate [11,16]. Other metabolites, such as cholesterol, bind to the outer hydrophobic surface of VDAC and interact with residue E73 of mVDAC1 [18]. Of note, E73 is a charged residue that protrudes from the outside of the barrel into the hydrophobic membrane, which is unusual. E73 is critical for VDAC’s dimerization and interaction with hexokinase-I [8,19]. It is possible that cholesterol binding could regulate these conformational changes by binding E73 [20]. In addition, cholesterol and other sterols can modulate VDAC gating, and VDAC can affect distribution of cholesterol in the outer mitochondrial membrane (OMM) [21,22]. Some metabolites, such as NADH and ATP, connect the glycolysis pathway in the cytosol to the oxidative phosphorylation pathway in the IMM. The integral role VDAC plays in energy metabolism and calcium homeostasis has made it a potential therapeutic target for diseases including cancer and neurodegeneration [23,24,25,26,27,28,29,30,31,32]. In addition, VDAC participates in a mitochondrial stress response by regulating the formation of the mitochondrial permeability transition pore (mPTP), which forms in the inner membrane due to extreme stressors such as high oxidative stress or high cytosolic calcium [33,34]. VDAC is nonessential to mPTP formation; however, studies on ischemia reperfusion injury suggest a dependence of VDAC on mPTP opening. For example, accumulation of VDAC1 was shown to increase mPTP opening [34].

The abundance of VDAC in the OMM has allowed for the application of a variety of experimental techniques, beginning with its identification in 1976 from *Paramecium tetraureliai* mitochondria [35]. Structures of VDAC1 determined by X-ray crystallography and nuclear magnetic resonance (NMR) show a 19-stranded β-barrel with β-strands running antiparallel except for β-strands 1 and 19, which are parallel. The β-barrel is ~30 Å in diameter with an N-terminal moiety that forms an α-helix [7,8,9,36]. All three structures position the α-helix in the barrel midway inside the pore [7,8,9]. The structures show hydrophobic residues that face the membrane and many positive hydrophilic residues facing the β-barrel lumen. The distribution of positive residues lining the pore has been regarded as the driving force for its anion selectivity in the open state (Figure 1) [7,8,9]. The importance of VDAC to the mitochondrial function has resulted in studies aimed at identifying the conformational changes that VDAC undergoes and the impact of these changes on gating, selectivity, and oligomerization. Outstanding questions on the relationship between VDAC’s electrophysiological properties and the observed conformational changes could provide insight into the regulatory mechanism for gating.

## 3. Structures of Mammalian VDAC Orthologs

The global features reported in the three structures of mammalian VDAC orthologs are the same, but minor differences exist at the N-terminus. These structures were solved by a combination of NMR and X-ray crystallography using protein expressed in *E. coli* inclusion bodies and refolded. The two structures derived from NMR give insight into the possible conformers of the β-barrel.

The first NMR structure is derived from human VDAC1 (hVDAC1) (Figure 2) (PDB ID: 2K4T). The structure shows β-barrel dimensions that are approximately 30 Å in height and 25Å in diameter. Of note, while the conformational ensemble shows consistency within the distances between each β-strand, the β-barrel shape adopted multiple conformations, possibly showing how the β-barrel could elongate and take on an elliptical shape [7]. In addition, the n-terminal α-helix is relatively short compared to the other two structures, comprising only residues 6–10. The structure shows missing assignments for regions of the n-terminus and the loops between β-strands. There is also an α-helix seen in the structure between the 18th and 19th strands that is not present in the other two structures. The n-terminal α-helix interacts with hydrophobic residues on strands β10 and β11 of the β-barrel [7].

The second structure of hVDAC1 was solved by combining NMR and X-ray crystallography data (Figure 2) (PDB ID: 2JK4) [9]. The β-barrel dimensions of the structure are 31 Å by 35 Å for the cytosolic side of the pore, with the narrowest diameter of the pore measuring 15 Å by 10 Å where the N-terminus fills the pore. Of note, the NMR conformational ensemble showed that strands β1-β4 are less stable than the other β-strands. This instability is attributed to residue E73, which faces the lipid bilayer and is important for VDAC oligomerization and protein–protein interactions (Table 2) [9,37]. The structure also shows a longer N-terminal α-helix in the pore. Acidic residues on the α-helix form contacts with hydrophilic residues on strands β12–β16 of the β-barrel. Like the hVDAC1 structure solved by NMR, there are missing assignments for residues on the β-strands. However, more residues of the N-terminus are resolved for this structure [9].

The third structure solved by X-ray crystallography was of murine VDAC1 (mVDAC1) (PDB ID: 3EMN) [8]. mVDAC1 has a 99 percent sequence identity with hVDAC1 except for an additional 13 residues for mVDAC1. This structure shows that the overall features are also similar between mVDAC1 and hVDAC1. The dimensions of the structure are 35 Å for the height and 40 Å for the width. The structure shows that the diameter of the pore measures 27 Å while the diameter of the narrowest point, which contains the α-helix, is 14 Å. The α-helix contacts strand β1 shows a break in secondary structure, resulting in two shorter α-helices [8]. This break is not seen in the other two structures.

The dimensions of the β-barrel are similar across the three initial structures of VDAC1, with the main differences observed in the secondary structure composition of the n-terminus (Figure 2).

## 4. Role of VDAC in Mitochondrial Protein Import

The fungal ortholog of VDAC, porin (Por1), has been studied in the context of its interactions with other OMM proteins. The similarity between the two proteins shows evolutionary conservation (Figure 3 and Figure 4). Functional orthologs, such as porin, can shed light on possible interactions of VDAC in humans with OMM protein machinery. For example, Por1 has been shown to interact with components of the translocase of the outer mitochondrial membrane (TOM) complex and other components of the mitochondrial protein import machinery [38]. Recent studies show that Por1 associates with Tom22, an α-helical receptor protein of the TOM complex. This sequestration is proposed to regulate the oligomerization of the TOM complex as Tom22 is crucial for the formation of higher order TOM oligomers, as well as to ensure proper integration of Tom22 into the mature TOM complex [39]. Por1, in concert with cell cycle-dependent phosphorylation of Tom6, a β-barrel-associated subunit of the TOM complex, sequesters Tom22. Tom6 undergoes phosphorylation in a cell cycle-dependent manner, which may regulate its interactions with Por1 [39]. Prior to the solution of the full TOM complex, Por1 was used as a homology model for the structure of Tom40, the β-barrel and central protein translocation pore of the TOM complex [40,41,42]. Recent structures reveal that Tom40, like Por1, is a 19-stranded β-barrel with an α-helical N-terminal domain located in the barrel pore. The structural similarities between the two proteins also result in some functional overlap as seen by the transport of metabolites by Tom40 when VDAC is depleted [43]. One study shows that expression of Tom40 is increased in Por1 deletion strains [43]. The functional overlap of Tom40 and VDAC can also be seen in Alzheimer’s disease. Components of the TOM complex are players in neurodegenerative disease states such as Alzheimer’s disease due to its import of amyloid beta-peptides [44]. Similarly, hVDAC1 interacts with and promotes entry of amyloid β aggregates [29]. Further studies on the interactions between the TOM complex and VDAC in disease states can help discern differences in their functions.

Por1 has also been implicated in mitochondrial protein import into the IMM. A recent study shows that Por1 depletion did not decrease the abundance of carrier protein precursors translocating across the OMM. Rather, the depletion led to decreased integration of the carrier proteins into the IMM due to interactions of Por1 with the Translocase of the Inner Membrane 22 (TIM22) complex, also known as the carrier translocase [45]. Whole exome sequencing revealed TIM22 to be involved in improper carrier protein translocation, resulting in mitochondrial myopathy [46]. Thus, understanding the relationship between VDAC and TIM22 in these disease states can reveal how VDAC modulates TIM22 function. An understanding of Por1 interactions with the TOM and TIM complexes can help elucidate VDAC’s role in regulating mitochondrial dysregulation via mitochondrial protein import in humans.

The TOM complex and the components of the Sorting and Assembly Machinery (SAM) complex are necessary for the biogenesis and the integration of VDAC into the OMM [47]. The depletion of Sam50, the β-barrel integrase of the SAM complex, led to improper integration and function of Por1 in the yeast. In human cell lines, the depletion of Sam50 led to decreased levels of VDAC in higher order complexes and decreased steady-state levels of the protein overall, possibly showing improper integration into the membrane [48]. In addition, the depletion of the TOM complex components, specifically Tom40 protein, led to a significant decrease in VDAC1 import into human mitochondria [48]. This demonstrates reliance on the TOM complex for VDAC import. Studies on the biogenesis of Por1 reveal the molecular consequences of import defects that can provide insight to the molecular pathology of human diseases. For example, improperly inserted VDAC can be fatal in diseases such as Parkinson’s disease [49,50]. Pathogenesis of Parkinson’s disease can progress if the impaired mitochondria are not cleared due to defective quality control mechanisms [49,50].

In addition to the interactions with the major mitochondrial protein translocation machinery, VDAC also interacts with the translocator protein (TSPO), an OMM protein involved in cholesterol transport. This interaction inhibits activation of PINK1/Parkin-dependent mitophagy [51]. The Pink1/Parkin-dependent mitophagy relies on depolarization of the IMM which leads to an accumulation of Pink1 on the OMM and results in the recruitment of the ubiquitin ligase Parkin. This then results in the broad ubiquitination of the OMM proteins, including VDAC, and subsequent degradation by the proteasome [52]. When TSPO is bound to VDAC1, reactive oxygen species (ROS) are released, resulting in a change in membrane potential that facilitates Pink1 kinase import and prevents recruitment of Parkin, a cytosolic E3 ubiquitin ligase which results in mitophagy. During Pink1/Parkin-dependent mitophagy, VDAC1 is tagged for degradation by ubiquitin [53]. While some level of TSPO and VDAC1 binding is needed to regulate the mitophagy pathway, overexpression of TSPO led to increased binding to VDAC1 which resulted in an accumulation of dysfunctional mitochondria in cancer cells [54]. These studies illustrate that proteins of the OMM, especially β-barrel proteins, need to coordinate effectively to maintain mitochondrial homeostasis.

## 5. Human VDAC Isoforms

There are three known isoforms of VDAC in humans: hVDAC1, hVDAC2, and hVDAC3. Although the structures of hVDAC2 and hVDAC3 have not been solved, the three human isoforms of VDAC are thought to share the same general structural features: a C-terminal 19-stranded β-barrel forming a pore in the OMM capable of metabolite transport, and an N-terminal α-helix residing inside the pore [55].

hVDAC1 is the most abundant isoform in most tissues, except for the testes where hVDAC2 and hVDAC3 are most abundant [36,56]. All three isoforms are highly expressed in the heart, kidney, skeletal muscle, and brain [57]. In rats, although there is high level expression of mVDAC1 and mVDAC2 proteins in the liver, it is relatively low when compared to other tissues [58]. While no structures are available for hVDAC2 or hVDAC3, sequence variations predict small structural differences, particularly in the N-terminal α-helix. In hVDAC2, the α-helix is 11 residues longer than in the other two human isoforms. hVDAC3 does not contain the bilayer-facing glutamate residue which is present in hVDAC1 and hVDAC2. This residue is important for dimer formation in acidic cellular environments; when the glutamate is protonated, VDAC has a higher affinity for dimerization and a higher order oligomerization, which suggests that hVDAC3 is unable to form homo-oligomers [37].

## 6. Role of the Closed VDAC Conformation in Disease

The closed VDAC conformation is preferred upon an increase in mitochondrial membrane potential. Many questions surround how an increased membrane potential can be created and sustained at the OMM. The Donnan potential, or the difference in ions and chemicals across a semipermeable membrane such as the OMM, could lead to VDAC closure. A Donnan potential of 43 mV was measured between the cytosol and inner membrane space, which shows potential for a closed VDAC conductance state [59]. However, this measurement may vary depending on the conditions of the cell. It is more likely that proteins and metabolites such as NADH bind to the inside of VDAC’s pore, modulating the membrane potential by promoting reversible VDAC closure [5]. This closure due to ligand binding is also more physiologically relevant in comparison to voltage-only dependent closure. Some metabolites and proteins have been shown to modulate the membrane potential, allowing for VDAC closure [60,61]. However, the identity or mechanism of the modulator protein has not been discerned so it may likely be a protein that is blocking the pore, leading to a reversible closed state. Closed VDAC channels are more permeable to calcium from the cytosol [17]. An increase in flux of calcium into the mitochondria is known to lead to apoptosis [62]. The importance of calcium to cellular function has resulted in studies on the role VDAC plays in calcium transport between the ER and mitochondria. VDAC has been found in ER-mitochondria contact sites known as mitochondria-associated membranes (MAMs). MAMs are essential for the calcium buffering capacity of the mitochondria. VDAC1 has shown to be the primary transporter of calcium across the OMM, so contacts between it and channels in the ER, such as the inositol trisphosphate receptor (IP_3_R) or glucose regulated protein 75 (Grp75), are key players in disease pathogenesis (Table 2) [6,63,64]. Proteins associated with MAMs regulate calcium release from the ER into the mitochondria via VDAC1 [65]. Calcium transport between the two organelles has been implicated in Friedreich’s ataxia (FA) [64]. By increasing contacts between VDAC1, the ER membrane components and mitochondrial protein frataxin demonstrated improved FA symptoms. Frataxin promotes the interaction between VDAC1 and IP_3_R resulting in increased MAM contact sites [64]. In addition to calcium flux within MAMs, calcium also promotes ROS generation. A study showed that the interaction of Mcl-1, a Bcl-2 family protein with VDAC1, promoted an increase in calcium uptake into the mitochondria. The increased uptake led to increased ROS levels, and subsequently increased cancer cell migration and invasion, as seen in lung cancer cells [66].

Numerous studies have shown interactions of various proteins with the N-terminal α-helix of hVDAC1, with some resulting in disease states. For example, the cytoskeleton protein tubulin binds the open state of hVDAC1, decreasing the flux of ATP across the membrane [73,74]. In a cell cycle-dependent manner, free tubulin is postulated to regulate the gating behavior of VDAC by interacting with the inside of the channel and blocking conductance of metabolites [73,74]. This tubulin-induced closed state is different from voltage-induced closed states as it occurs at lower membrane potentials, and leads to one closed state in comparison to multiple closed states observed in voltage-induced states [14]. A cation selectivity and low ATP passage through the pore demonstrate similarities between both types of closed states [77]. Tubulin sensitivity varies across the three isoforms of VDAC [13]. hVDAC3 has lower sensitivity toward tubulin in comparison to both hVDAC1 and hVDAC2. Erastin, a small molecule that is capable of inducing apoptosis and has been studied in oncotherapy, reverses tubulin blockage of VDAC, allowing for cells to use oxidative phosphorylation for metabolism rather than on aerobic glycolysis alone [78]. Reliance on aerobic glycolysis alone is seen in cancerous cells. Different isoforms of human tubulin, for example, tubulin βII and βIII, have been shown to associate with the OMM and block the pore of VDAC [79,80]. Blockage of VDAC by tubulin βII is important in cardiomyocytes where ATP/ADP is compartmentalized near complexes. For example, decreased diffusion of ADP through VDAC due to blockage by tubulin βII affects the complex formed by the mitochondrial creatine kinase (MtCK), which associates to the outer surface of the IMM and the adenine nucleotide translocase (ANT), located in the inner membrane. Decreased ADP diffusion results in increased functional coupling with MtCK and ANT. This increase in coupling leads to more targeting of ADP to ANT, stimulating oxidative phosphorylation. Interactions of VDAC with tubulin βII influence the bioenergetics of the mitochondria [81]. Changes in tubulin βII blockage of VDAC is associated with cardiac ischemia–reperfusion injury [82]. Other cytoskeleton components that associate with VDAC include plectin 1b, an isoform of the intermediate filament protein known as plectin [83]. Plectin 1b also plays a role in mitochondrial respiration. For example, when plectin 1b is knocked out, there is an observed decrease in the apparent K_m_ of ADP, indicating an increase in VDAC due to increased permeabilization of ADP [83].

VDAC also undergoes post-translational modifications. hVDAC1 is phosphorylated by a protein kinase C (PKC), a cAMP-dependent protein kinase A (PKA), and a glycogen synthase kinase-3β (GSK3β) [84]. Free tubulin, subunits of microtubules located in the cytosol, can block VDAC and decrease the permeability of metabolites such as ATP. Interestingly, a study shows that VDAC phosphorylation by PKA increased tubulin blockage of VDAC [12]. VDAC could play a role in cancer cells that are characterized by disintegrated microtubule networks, possibly altering the permeability of metabolites such as ATP/ADP [85].

Alpha-synuclein is another molecule that binds to open state VDAC and induces a partial and reversible closed state at low physiological membrane potentials [86]. Alpha-synuclein can also translocate through the VDAC at higher membrane potentials and target respiratory complexes after entering the inner membrane space. The binding of alpha-synuclein to VDAC also allows for more cationic molecules, such as calcium, to traverse the channel [87]. This increased net flux is seen in disease states such as Parkinson’s disease. Another metabolite that can bind within the open pore is β-NADH [7,10]. β-NADH decreases the permeability of the outer membrane to ADP, possibly by promoting VDAC gating (Figure 4) [71]. hVDAC1 residues that interact with β-NADH include G242, L243, I244, A261, L263, and D264, on β-strands 17 and 18 [7]. β-NADH levels increase in anoxic conditions, stabilizing the closed state of VDAC. The stabilized closed state allows calcium ions to migrate through the channel which may lead to apoptosis [67]. Hexokinase II, a central component of the glycolytic pathway, is overexpressed in tumor cells, and, when bound to VDAC1, it lowers OMM permeability and increases glycolysis, thereby aiding in tumorigenesis [75,76]. When VDAC is blocked by hexokinase, a key metabolite needed for oxidative phosphorylation, ATP cannot pass through the channels. In healthy cells, most of the ATP is generated from mitochondrial oxidative phosphorylation; however, during tumorigenesis, cancerous cells utilize ATP generated from aerobic glycolysis. This switch to increased glycolysis in cancer cells is known as the Warburg phenomenon [88]. Glycolysis produces carbon sources necessary for the biosynthesis of nucleotides, lipids, and amino acids, which is vital for tumorigenesis [89]. The role of hexokinase in disease states is also seen in bacterial infection. *Chlamydia trachomatis*, a human obligate intracellular bacterial pathogen, induces hexokinase II binding to VDAC channels which leads to increased bacterial survival as this binding decreases apoptosis and increases infected cell proliferation [76].

Under stress, VDAC also forms homo-oligomers and hetero-oligomers. It is thought that residues on β-strands 1, 2, 18, and, 19 form an interface for oligomerization (Figure 5) [9]. Mutational analysis and cross-linking studies have shown that cysteine residues, C127 and C232 of rat VDAC1, help to form higher order oligomers (Figure 5). The roles of these two cysteines, located on β-strands 8 and 16, respectively, in addition to residues predicted to require higher energy for insertion into a lipid bilayer, such as A231 on β-strand 16, T116 on β-strand 7, and G140 on β-strand 9, suggest that residues on other β-strands may also play a role in VDAC1 oligomerization. These stabilized VDAC oligomers are also able to release mtDNA into the cytosol under mild stress, which leads to autoimmune responses as seen in lupus [68]. Thus, the oligomeric states of VDAC have been implicated in the promotion of autoimmune diseases such as systemic lupus erythematosus (SLE) and in Parkinson’s disease (PD) [68]. Inhibition of the oligomerization of VDAC1 was shown to decrease lupus-like symptoms in mice [68]. In addition, VDAC oligomers have been extensively studied in relation to cell apoptosis. For example, many studies have correlated the release of cytochrome c with VDAC oligomer formation [15]. Cytochrome c, upon release into the cytosol, interacts with pro-apoptotic proteins, leading to cell death [69].

## 7. Conclusions

VDAC is a key player in many mitochondrial processes such as signaling, apoptosis, and calcium homeostasis. Studies of VDAC over the last fifty years have provided remarkable insight into the structure and function of VDAC under various cellular conditions. Many questions remain, such as how the oligomeric states of VDAC contribute to its multifunctional nature, and whether the conformational states of VDAC in these instances matter to the structure of the closed states, the structure of VDAC oligomers, the role of the N-terminal α-helix, and VDAC’s role in apoptosis and MOMP. A better understanding of the structure and function of VDAC during disease states can aid in the design therapeutics that modulate VDAC’s conductance or binding partners in a specific manner. This specificity can allow for better therapeutics and innovation in areas such as neurodegenerative diseases, cancer, and cardiac treatments. The importance of VDAC to mitochondrial homeostasis makes it imperative to understand the structure–function relationship of the various states of VDAC and their roles in disease.

## Figures and Tables

**Figure 1 cells-10-01737-f001:**
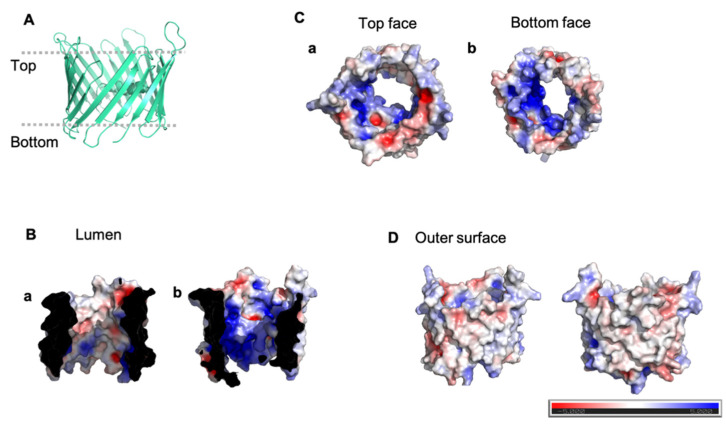
Electrostatic distribution of VDAC. Electrostatic distribution inside the lumen, on the outside surface of hVDAC1 (PDB ID: 2JK4), and top and bottom of the pore. (**A**) Shows cartoon representation of hVDAC1. (**B**) shows the inside of the lumen and the N-terminal helix (a) and the other side of the lumen (b). (**C**) shows the top face of VDAC (a) and bottom face (b). (**D**) Shows the outside surfaces. Generated using the APBS plugin in Pymol.

**Figure 2 cells-10-01737-f002:**
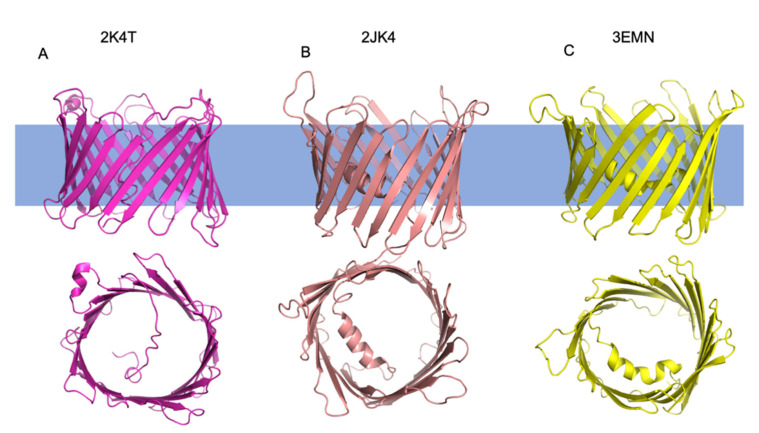
Structures of Mammalian VDAC1 orthologs. Cartoon representation of structures solved in 2008 by three separate groups. (**A**) Shown in pink are the side and top (according to orientation shown) views of hVDAC1 solved by NMR (PDB ID: 2K4T). (**B**) Shown in salmon are the side and top views of hVDAC1 solved by NMR and X-ray crystallography (PDB ID: 2JK4). (**C**) Shown in yellow are the side and top views of mVDAC1 solved by X-ray crystallography (PDB ID: 3EMN).

**Figure 3 cells-10-01737-f003:**
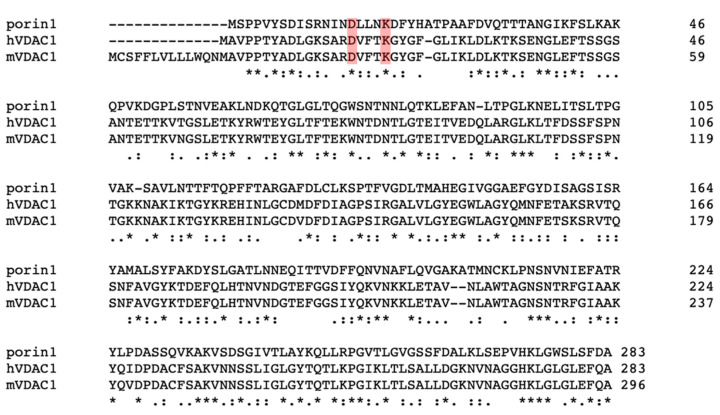
Sequence alignment between yeast porin 1, mVDAC1, and human VDAC1. Although there is not a high sequence identity between the hVDAC1 and yeast por1, some residues are similar, including D15 and K19 of porin, which were mutated in studies such as in Blachly-Dyson, Peng, Colombini, & Forte, 1990. Sequence alignment was performed using Clustal Omega.

**Figure 4 cells-10-01737-f004:**
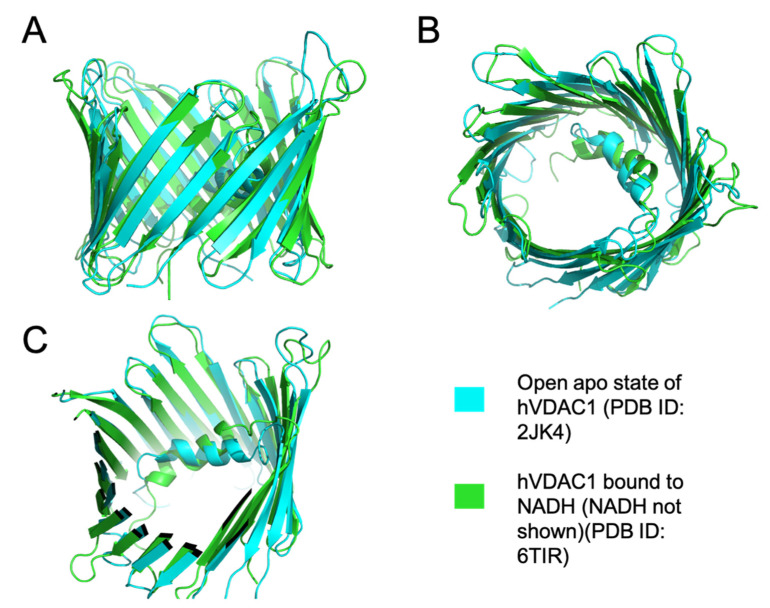
Superimposition of open apo hVDAC1 and NADH- hVDAC1 structures. (**A**) Shows the side view of the superimposed structures of the high and low conducting states of hVDAC1 (PDB ID: 6TIR and 2JK4). (**B**) Shows the top view of the aligned structures. (**C**) Shows a clipped structure of hVDAC1 to show the N terminal α-helix. The structures were aligned and depicted using Pymol.

**Figure 5 cells-10-01737-f005:**
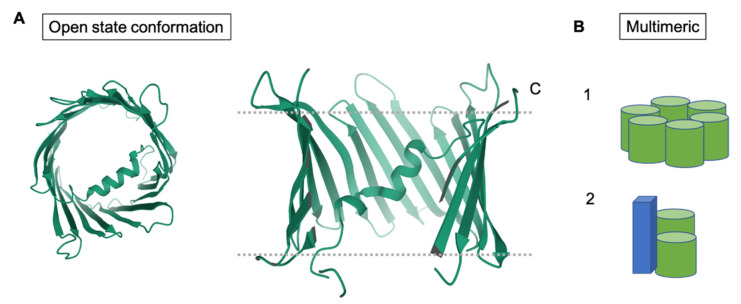
Structural conformations of VDAC. (**A**) Shows top and side views of hVDAC1 in the open conformation (PDB ID: 2JK4). (**B**) Models of the oligomeric structures of VDAC, including the homomeric and hexameric forms. The green cylinders represent VDAC. The blue cuboid represents a protein, other than VDAC, that associates to form a hexameric complex.

**Table 1 cells-10-01737-t001:** Our current understanding of structural conformations and their relationships to gating and selectivity of anions or cations.

Gating	Possible Structural Conformations	Specificity
Open	α-helix inside pore Circular β-barrel shape	Higher anion/cation ratio
Closed	α-helix stays inside pore or there is movement of α-helix within pore Elliptical β-barrel shape Movement of β-strands away from β-barrel wall	Lower anion/cation ratio
Possibly closed	Oligomeric structures including homo-oligomeric and hetero-oligomeric	Possible release of higher molecular weight molecules than ions, ATP, NADH etc.
Closed	Protein blockage of pore No other structural changes	Lower anion/cation ratio

**Table 2 cells-10-01737-t002:** The interactome of VDAC in various states along with the associated diseases. The gating of VDAC includes: open, closed, multimeric, and blockage of pore by specific proteins and metabolites. When these interactions are disrupted, they lead to disease states, listed in the “Disease States” row.

Gating of VDAC	Molecules that Bind to Each State of VDAC	Disease States
Open	ADP/ATP [35] Cl- [35] NADH/NAD+ [7,10]	NADH increases during anoxia and leads to VDAC closure to ATP [67]
Closed	Calcium [17] ○ Mitochondria-Associated Membranes (MAMS) [64] ○ IP3R [63], ○ Grp75 [63] Mcl-1 [66]	Increase in calcium into the mitochondria leads to apoptosis [62] Friedreich’s ataxia (FA) with dysregulation of MAMs [64] Lung cancer cell migration due to decreased calcium uptake with binding to Mcl-1 [66]
Multimeric	Mitochondrial DNA [68] Cytochrome C [69] Bcl-XL [70] BAX [71] BAK [71]	Parkinson’s disease due to oligomerization [68] Systemic lupus erythematosus (SLE) due to mtDNA release into cytosol [68] Apoptosis due to cytochrome C release into cytosol [15] Association to Bcl-XL as well as other proteins leads to disease states such as ischemia reperfusion injury [70,72]
Blockage of Pore	Free Tubulin [73,74] Alpha-synuclein [63] Hexokinase II [75]	Antagonism of Tubulin’s association with VDAC leads to Cancerogenesis and cell proliferation [73,74] Alpha-synuclein, the Parkinson’s disease associated protein can cause dysregulation in calcium homeostasis. [72] Growth of tumor cells due to hexokinase binding to hVDAC1 [76] Bacterial infections such as Chlamydia trachomatis due to hexokinase association with hVDAC1 [76]

## Data Availability

No new data were created or analyzed in this study. Data sharing is not applicable to this review article.

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
