# Peer review of "The Role of Voltage-Dependent Anion Channel in Mitochondrial Dysfunction and Human Disease"

_cells, 2021, doi:10.3390/cells10071737_

Round 1
Reviewer 1 Report
This is an interesting and detailed review that shows the structure of the voltage dependent anion channel and its role in physiological and pathological processes. I think that authors need to adjust the work before publishing.
First, the authors write that «In healthy cells, most of the ATP is generated from mitochondrial oxidative phosphorylation however, during tumorigenesis, cancerous cells utilize ATP generated from aerobic glycolysis, and this switch in the mechanism of energy production in cancer cells is known as the Warburg phenomenon» (lines 355-359).
This is a very important statement, and the Authors have to better describe this metabolic reprogramming.
It should be remembered that the occurrence of this metabolic shift in tumours was initially suggested in 1923 by Otto Warburg, and in recent years the “Warburg effect” has been revisited and heavily investigated using advanced approaches (transcriptomics, proteomics, metabolomics, etc). Many studies showed that increased glycolysis is not relevant for energy production, as Warburg originally predicted, but rather serves to provide building blocks of macromolecules for biosynthetic processes that are critical for enhanced tumor growth (Koppenol et al 2011 Nat. Rev. Cancer and others).
In addition, the authors should also mention the possible role of VDAC in the regulation of mitochondrial calcium-dependent MPT pore opening (PMID: 28713289), this will enrich the work.
Author Response
This is an interesting and detailed review that shows the structure of the voltage dependent anion channel and its role in physiological and pathological processes. I think that authors need to adjust the work before publishing.
First, the authors write that «In healthy cells, most of the ATP is generated from mitochondrial oxidative phosphorylation however, during tumorigenesis, cancerous cells utilize ATP generated from aerobic glycolysis, and this switch in the mechanism of energy production in cancer cells is known as the Warburg phenomenon» (lines 355-359).
This is a very important statement, and the Authors have to better describe this metabolic reprogramming.
It should be remembered that the occurrence of this metabolic shift in tumours was initially suggested in 1923 by Otto Warburg, and in recent years the “Warburg effect” has been revisited and heavily investigated using advanced approaches (transcriptomics, proteomics, metabolomics, etc). Many studies showed that increased glycolysis is not relevant for energy production, as Warburg originally predicted, but rather serves to provide building blocks of macromolecules for biosynthetic processes that are critical for enhanced tumor growth (Koppenol et al 2011 Nat. Rev. Cancer and others).
In addition, the authors should also mention the possible role of VDAC in the regulation of mitochondrial calcium-dependent MPT pore opening (PMID: 28713289), this will enrich the work.
Author response:
1) In Line 379, “energy production” was changed to “increased glycolysis”. In addition a sentence was added describing how increased glycolysis is critical for tumorigenesis in Lines 378-381. The role of increased glycolysis in anabolic reactions was highlighted.
2)The role of VDAC in the regulation of MPT pore opening as well as its role in disease were added in lines 96-100.
Reviewer 2 Report
Numerous metabolites, such as respiratory substrates, ADP and Pi enter mitochondria through VDAC. On the other side, high-energy phosphates - ATP and phosphocreatine are channeled out from mitochondria through the VDAC to drive the cellular energy metabolism. Control of the energy fluxes through VDAC is tightly regulated by different mechanisms. In addition, VDAC represents a channel for the release of several factors from mitochondria in response to apoptotic stimuli. VDAC and adenine nucleotide translocase (ANT) were identified as proteins involved in the mitochondrial permeability transition pore complex (MPTP).
This review rather comprehensively describes VDAC function, structure and changes in diseases. Nevertheless, some important aspects of VDAC regulations are missing.
1) Although potential role of free tubulin addition in the VDAC transitions has been briefly described (works of Rostovseva and others), the fact that other cytoskeletal proteins can control VDAC permeability and therefore cellular energy fluxes was not sufficiently mentioned. In particular, the important role of tubulin beta-II isoform in VDAC and mitochondrial regulations should be added. Since tubulin beta-II bound to VDAC can trigger its transition to closed state, regulating mitochondrial ADP/ATP fluxes and thus overall cellular bioenergetics.
2) A possible role of the specific plectin isoforms in VDAC regulation can also be mentioned.
3) Moreover, the closed VDAC creates a particular condition for the metabolic, ATP/ADP micro-compartmentation and mitochondrial creatine kinase (mitCK) coupling with ANT in cardiac cells. The coupled mitCK is a key element of creatine-phosphocreatine shuttle for the effective intracellular energy transfer. These phenomena can be cited in the manuscript.
Author Response
Numerous metabolites, such as respiratory substrates, ADP and Pi enter mitochondria through VDAC. On the other side, high-energy phosphates - ATP and phosphocreatine are channeled out from mitochondria through the VDAC to drive the cellular energy metabolism. Control of the energy fluxes through VDAC is tightly regulated by different mechanisms. In addition, VDAC represents a channel for the release of several factors from mitochondria in response to apoptotic stimuli. VDAC and adenine nucleotide translocase (ANT) were identified as proteins involved in the mitochondrial permeability transition pore complex (MPTP).
This review rather comprehensively describes VDAC function, structure and changes in diseases. Nevertheless, some important aspects of VDAC regulations are missing.
1) Although potential role of free tubulin addition in the VDAC transitions has been briefly described (works of Rostovseva and others), the fact that other cytoskeletal proteins can control VDAC permeability and therefore cellular energy fluxes was not sufficiently mentioned. In particular, the important role of tubulin beta-II isoform in VDAC and mitochondrial regulations should be added. Since tubulin beta-II bound to VDAC can trigger its transition to closed state, regulating mitochondrial ADP/ATP fluxes and thus overall cellular bioenergetics.
2) A possible role of the specific plectin isoforms in VDAC regulation can also be mentioned.
3) Moreover, the closed VDAC creates a particular condition for the metabolic, ATP/ADP micro-compartmentation and mitochondrial creatine kinase (mitCK) coupling with ANT in cardiac cells. The coupled mitCK is a key element of creatine-phosphocreatine shuttle for the effective intracellular energy transfer. These phenomena can be cited in the manuscript.
Author response:
Lines 329-344 have been added to reflect the reviewer’s comments.
The binding of different isoforms of tubulin to VDAC is mentioned.
In addition, the role of that interaction in relation to mtCK-ANT coupling is elaborated on.
The possible role of plectin 1b is also briefly mentioned.
Round 2
Reviewer 1 Report
I have no more comments.